# Longitudinal Genome-Wide Association Study of Cognitive Impairment after Subarachnoid Hemorrhage

**DOI:** 10.3390/biomedicines12071387

**Published:** 2024-06-22

**Authors:** Eun Pyo Hong, Seung Hyuk Lim, Dong Hyuk Youn, Sung Woo Han, Harry Jung, Jae Jun Lee, Jin Pyeong Jeon

**Affiliations:** 1Institute of New Frontier Research, Hallym University College of Medicine, Chuncheon 24254, Republic of Korea; ephong0305@hallym.ac.kr (E.P.H.); ephong0305@gmail.com (S.H.L.); zk61326@hallym.ac.kr (D.H.Y.); hansw29@hallym.ac.kr (S.W.H.); harry@hallym.ac.kr (H.J.); 2Department of Anesthesiology and Pain Medicine, Hallym University College of Medicine, Chuncheon 24253, Republic of Korea; 3Department of Neurosurgery, Hallym University College of Medicine, 77 Sakju-ro, Chuncheon 24253, Republic of Korea

**Keywords:** cognitive impairment, longitudinal genome-wide association study, polygenic risk score, subarachnoid haemorrhage

## Abstract

Objectives: The occurrence of cognitive deficits after subarachnoid hemorrhage (SAH) is highly possible, leading to vascular dementia. We performed a novel longitudinal genome-wide association study (GWAS) to identify genetic modifications associated with cognitive impairment following SAH in a long-term prospective cohort study. Materials and Methods: This GWAS involved 153 patients with SAH sharing 5,971,372 markers after high-throughput imputation. Genome-wide Cox proportional hazard regression testing was performed to estimate the hazard ratio (HR) and 95% confidence interval (CI). Subsequently, a weighted polygenetic risk score (wPRS) was determined, based on GWAS-driven loci and risk stratification. Results: Cognitive impairment was observed in 65 patients (42.5%) during a mean follow-up of 37.7 ± 12.4 months. Five genome-wide signals, including rs138753053 (*PDCD6IP*-*LOC101928135*, HR = 28.33, *p* = 3.4 × 10^−8^), rs56823384 (*LINC00499*, HR = 12.47, *p* = 2.8 × 10^−9^), rs145397166 (*CASC15*, HR = 11.16, *p* = 1.7 × 10^−8^), rs10503670 (*LPL*-*SLC18A1*, HR = 2.88, *p* = 4.0 × 10^−8^), and rs76507772 (*IRS2*, HR = 5.99, *p* = 3.5 × 10^−8^), were significantly associated with cognitive impairment following SAH. In addition, the well-constructed wPRS containing five markers showed nominal ability to predict cognitive impairment (AUROC = 0.745, 95% CI: 0.667–0.824). Tertile stratification showed a higher effectiveness in predicting cognitive impairment, especially in those with haptoglobin 2-1 (HR = 44.59, 95% CI: 8.61–231.08). Conclusions: Our study revealed novel susceptible loci for cognitive impairment, longitudinally measured in patients with SAH. The clinical utility of these loci will be evaluated in further follow-up studies.

## 1. Introduction

Subarachnoid hemorrhage (SAH) due to intracranial aneurysm (IA) rupture is a life-threatening condition, with high mortality (up to 50%) and morbidity in the survivors (up to 30%) [1,2]. Accordingly, clinical studies have focused on improving survival by investigating an efficient method to secure the ruptured aneurysm (surgical clipping vs. endovascular coil embolization), as well as early diagnosis and treatment of delayed cerebral ischemia (DCI) and hydrocephalus. Due to these efforts and improvements in emergency medical systems, mortality and morbidity due to SAH have improved somewhat over the past 20 years [3]. As survival rates have increased, interest in cognitive impairment has grown in the clinical field. SAH occurs at a relatively young age compared to neurodegenerative diseases or ischemic strokes. The most common age of SAH occurrence is in the mid-40 s to mid-60 s [4]. An incidence rate of cognitive impairment up to 50% following SAH has been reported [5,6]. Compared to cognitive impairment associated with Alzheimer’s disease (AD) and ischemic stroke, studies on cognitive impairment after SAH are relatively lacking. In particular, genetic studies are very scarce, and most of the studies conducted have investigated the association with apolipoprotein (*APOE*) in a small number of patients. In an analysis of 46 SAH patients, Louko et al. reported that *APOE ɛ4* carriers exhibited difficulties in visual memory tasks and color naming [7]. Lanterna et al. also reported that the *APOE ɛ4* allele was associated with cognitive impairment, cognitive functional outcomes, and delayed ischemic neurologic deficits according to the Mini-Mental State Examination (MMSE) at least 6 months after ictus in 101 patients with SAH [8]. Beyond the association between *APOE* epsilon alleles and cognitive impairment, Han et al. recently reported that haptoglobin (Hp) binding freely to hemoglobin was closely associated with cognitive impairment following SAH [9]. Interestingly, patients experiencing SAH with Hp2-2 exhibited a significantly higher chance of 6-month or long-term cognitive impairments than those with Hp1-1 (*p* < 0.01) [9].

Individual risks associated with a disease phenotype over a certain time period can be assessed by summing the genetic effects [10]. Considering that there are multiple genes related to cognitive function, a longitudinal genome-wide association study (GWAS) is informative in evaluating cognitive impairment in patients with SAH. GWASs are designed to identify phenotype-associated single nucleotide polymorphisms (SNPs) by comparing allele frequencies at the level of the whole genome. To the best of our knowledge, there have been few studies on cognitive impairment after SAH based on a GWAS. Thus, we performed a longitudinal GWAS for the first time to identify candidate variants associated with cognitive impairment in patients following SAH.

## 2. Material and Methods

### 2.1. Study Population

This study involved patients with SAH in the database of the study entitled “The First Korean Stroke Genetics Association Research” initiative beginning in May 2015 [9]. This was a prospective multicenter observational study that evaluated the prognosis of genome-based cerebrovascular disease patients, including those with SAHs [9,11,12,13]. From the database, we selected SAH patients with the following conditions: (1) adult patients more than 18 years of age, (2) the presence of SAH due to aneurysm rupture, (3) patients possessing GWAS results and cognition tests, and 4) patients with follow-up at least 6 months after ictus. The exclusion criteria were: (1) SAH due to trauma or infection, (2) patients without GWAS results, (3) insufficient records, including a lack of radiological findings and cognition tests, (4) follow-up loss, and (5) those who could not perform cognitive function tests (Appendix A). All methods were carried out in accordance with relevant guidelines and regulations. All protocols of this study, including obtaining written informed consent from all patients, were approved by the Institutional Review Board and Ethics Committee of the Hallym University Chuncheon Sacred Heart Hospital (No. 2016-3, 2019-06-006-012, and HIRB-2022-042).

### 2.2. Study Outcomes

The primary outcome was the identification of variants associated with cognitive impairment using a longitudinal GWAS following SAH. The secondary outcome was the evaluation of the individual polygenic risk of cognitive impairment after SAH based on haptoglobin subtypes—both of which have been suggested as risk factors [8,9]—by comparing the summed risk allele count per subject based on GWAS-driven SNPs. We performed an unweighted or weighted polygenic risk score (uwPRS or wPRS) assessment to predict the events of cognitive impairment after SAH, based on GWAS-driven susceptible loci. Cognitive function tests were conducted using the Korean version of the Mini-Mental State Examination (K-MMSE) (Appendix A). The test was routinely performed 6 months after SAH ictus and was performed regularly every year thereafter. An MMSE score of less than 27 was defined as cognitive impairment [14,15,16]. Three types of haptoglobin were classified, and then Western blotting was performed, again targeting only Hp2-1. For analysis, a 1:75 dilution of serum was obtained by adding 1 μL of serum to 74 μL of phosphate-buffered saline. Samples were prepared by mixing a serum diluent with an equal volume of 2x-SDS sample buffer (Bio-Rad, Hercules, CA, USA) and then boiling the mixture at 95 °C for 8 min. After boiling, 10 μL of the sample was loaded on 15% polyacrylamide gel and electrophoresed for 150 min at 100 V (Bio-Rad, CA, USA). Detailed information regarding Hp phenotypes is shown elsewhere (Appendix A) [9,17].

### 2.3. Genotyping and Quality Controls

Genomic DNA derived from the peripheral blood of SAH patients who were genotyped by the Axiom^TH^ Asian Precision Medicine Research Array (APMRA) (Thermo Fisher Scientific, Waltham, MA, USA) was used. High-quality plates exhibited a plate pass rate of >95% for the samples and an average call-rate of the passed samples of >99%. Out of 802,688 SNPs, 512,503 SNPs passed the quality control filters (i.e., genotyping call rate ≥ 95%, minor allele frequency ≥ 1%, and Hardy–Weinberg equilibrium of *p* ≥ 1 × 10^−6^).

### 2.4. High-Throughput Imputation and Quality Control

We used the Eagle v2.4 and Minimac4 programs to pre-phase the genotypes of individual SNPs and impute the large amount of SNPs and missing genotype values for 153 subjects with 490,038 SNPs. We applied an Asian-specific reference panel (GRCh37/hg19) generated by the GenomeAsia 100K Project, which was supported by the National Institutes of Health (NIH) National Heart, Lung, and Blood Institute (NHLBI) [18]. A high-throughput imputation was performed in the Michigan Imputation Server v1.5.7 (https://imputationserver.sph.umich.edu/index.html#!run/minimac4, accessed on 5 January 2023). Out of 292,174,934 sites including monomorphic regions, 21,494,648 sites with imputation score *R*^2^ thresholds above 0.5 remained in 153 patients with SAHs. Finally, 5,971,372 SNPs were analyzed in this longitudinal GWAS, which passed the quality control tests with a genotyping call rate ≥ 95%, a minor allele frequency ≥ 1%, and a Hardy–Weinberg equilibrium of *p* ≥ 0.001.

### 2.5. Statistical Analysis

The discrete variables were presented as the number of subjects and the percentage. Continuous variables were presented as the mean and standard deviation (SD). Univariate and multivariate analyses were performed in the Cox proportional hazard (CPH) regression model to estimate the hazard ratio (HR) and 95% confidence interval (CI) using STATA software v17.0 (Stata Corp., College Station, TX, USA). Principal component analysis (PCA) was performed to evaluate sample heterogeneity and distribution based on 153 SAH patients and 2504 subjects of the 1000 Genome (1KG) reference panel (Phase 3, version 5). The PCA values were transformed using four multidimensional scales (MDSs). GWAS-driven CPH regression analysis was conducted based on cognitive impairment during the follow-up period after adjusting for gender, age, and four genetic ancestry MDSs. Additional GWAS subgroup analyses were performed based on Hp phenotypes, which are known risk factors for cognitive impairment following SAH [9,19]. This longitudinal GWAS were performed after adjustment for the following covariates, such as age, gender, and four PCA values (model 1), plus a history of diseases (i.e., hypertension, type 2 diabetes, hyperlipidemia) and/or smoking status (model 2 or 3), by using the “*survival*” library of R package software implemented by the “*coxph*” function (https://cran.r-project.org/web/packages/survival/index.html, accessed on 5 January 2023). Manhattan and regional associations were drawn by the “*qqman*” in R package v3.6.2 (https://cran.r-project.org/web/packages/qqman, accessed on 5 January 2023) and LocusZoom v1.4, written in modified Python v.2.7.5 and R scripts (http://locuszoom.org/, accessed on 5 January 2023). Further, the quantile–quantile (Q–Q) plot was illustrated based on the current GWAS data containing 5,976,620 genotyped-imputed SNPs in 153 patients with SAH by using the library “*qqplot*” of R package, and the genomic inflation factor (λgc) was then calculated based on median Chi-squared statistics using “*QQperm*” of the R package with the statistics of the observed *p*-values in the current longitudinal GWAS.

Subsequently, the uwPRS and wPRS were evaluated based on GWAS-driven loci to determine whether this risk-scoring system is more suitable for predicting cognitive impairment after SAH compared to the method involving the risk association of individual SNPs. The uwPRS value of each SNP was summed by the number of risk alleles (i.e., 0, 1, or 2 copies) in the GWAS-driven SNPs (*p* < 5 × 10^−8^). The final wPRS model was constructed by summing the weighted genotypes of individual SNPs and multiplying the log-transformed HR to normalize a distribution of summed risk alleles. The predictability, sensitivity, and specificity of the constructed model was evaluated by the area under the receiver operating curve (AUROC), while using the “*roctab*” and “*roccomp*” packages of STATA v17.0 to estimate a summary statistic of AUCROC and compare its outcomes. A further association between risk groups was tested in either uncategorized PRS or categorized PRS after stratifying them into tertiles (T1, low risk; T2, medium risk; and T3, high risk), while adjusting for covariates. Tertiles were automatically clustered by using the “*xtile*” package with the option “*nquantiles(3)*” packages of STATA software v17.0 to evaluate the risk of developing incidental events of cognitive impairment after SAH.

## 3. Results

### 3.1. Baseline Characteristics of the Study Participants

The clinical characteristics of the enrolled patients with SAH are described in Table 1. After excluding patients who were not eligible for the study, a total of 153 were included in the final analysis (Appendix A). During a mean follow-up of 37.7 ± 12.4 months, cognitive impairment was observed in 65 surviving patients (42.5%) who underwent MMSE. Compared to SAH patients without cognitive impairment, those with cognitive impairment were older (*p* < 0.001) and exhibited increased hypertension (*p* = 0.002). Diabetes mellitus was more common in patients with cognitive impairment, but it was not statistically significant (*p* = 0.085). Between the two groups, there were no significant differences in SAH-related variables such as aneurysm size, location, higher Hunt and Hess grade, higher Fisher grade, delayed cerebral ischemia, and the presence of hydrocephalus with a shunt. Most patients (n = 139, 90.8%) enrolled in the study underwent coil embolization. We further included information regarding the frequency of Hp subtypes for both the cognitive impairment and non-cognitive impairment of patients with SAH (Appendix A). Regarding Hp subtypes, patients with Hp2-2 showed a higher chance of cognitive impairment than those with Hp1-1 (HR = 9.82, 95% CI: 1.35–71.30; *p* = 0.024).

### 3.2. Longitudinal Genome-Wide Association Study

There existed no ethnic or sample heterogeneity in the integrated dataset including the 1KG samples (Figure 1A) and 153 SAH patients (Figure 1B) according to PCA. According to the Manhattan plot for the longitudinal GWAS in the CPH model, five SNPs showed genome-wide significance (*p* < 5 × 10^−8^), and these were associated with cognitive impairment following SAH (Figure 1C and Table 2). The value of the genomic inflation factor (λgc) was 1.107 for the current GWAS, as illustrated in the Q–Q plot (Figure 1D). Out of five SNPs, rs56823384 (*LINC00499*) and rs145397166 (*CASC15*) were imputed from the Asian-specific reference panel (GenomeAsia 100K Project) (Table 2). Among them, four showed a single point variation (Appendix A), and the other was a haplotype-tagged SNP (Figure 2). Although the intronic SNP of rs56823384 (ncRNA, *LINC00499*) showed the most significant association with cognitive impairment (HR = 12.47, 95% CI: 5.42–28.67; *p* = 2.8 × 10^−9^), there was no other strong LD block in the 4q28.3 loci. Likewise, three other SNPs, rs138753053 (*PDCD6IP*-*LOC101928135*, 3p22.3), rs145397166 (*CASC15*, 6p22.3), and rs10503670 (*LPL*-*SLC18A1*, 8p21.3), did not exhibit additional tagging SNPs within ±500 kb in the associated regions (Appendix A). Interestingly, the rs76507772 SNP near the *IRS2* gene showed a complete LD with four other downstream SNPs, including rs75291961, rs1865433, rs80076726, and rs113236650 (HR = 5.99; 95% CI: 3.17–11.32; *p* = 3.5 × 10^−8^) (Figure 2A). Cognitive impairment developed more rapidly in SAH patients with AC heterozygous genotypes than those with AA homozygous genotypes (Figure 2B). The association between cognitive impairment and the five SNPs remained significant in multiple CPH regression models after adjusting for clinical covariates (Table 3).

### 3.3. Subgroup Analysis after Stratification into Hp Subtypes

Five SNPs, including rs138753053, rs56823384, rs145397166, rs10503670, and rs76507772, showed suggestive or nominally significant associations for developing incidental events of cognitive impairment after SAH cognitive impairment in patients with Hp2-1 and Hp2-2 (5 × 10^−7^ < *p* < 0.011) (Appendix A).

### 3.4. Polygenic Risk Assessment of Cognitive Impairment

In the subsequent analysis, the uwPRS and wPRS predicting the development of cognitive impairment events after SAH were proposed, based on an allele scoring system of the five genome-wide significant SNPs (Figure 3 and Appendix A). A combined uwPRS exhibited moderate accuracy in predicting an incidental event of cognitive impairment in patients with SAH (AUROC = 0.739, 95% CI: 0.659–0.817). The wPRS slightly improved the prediction accuracy (AUROC = 0.745, 95% CI: 0.667–0.824) (Figure 3A). The two models showed no difference in AUCROC outcomes (Chi-square = 1.63, *p* = 0.2018). The original (uncategorized) wPRS model showed a statistically significant association with developing incidental events of cognitive impairment after SAH (HR = 1.23, 95% CI: 1.16–1.29; *p* = 2.1 × 10^−4^), compared to the uwPRS model (HR = 1.45, 95% CI: 1.31–1.59; *p* = 8.4 × 10^−14^). The pattern of predictability in either the single uwPRS or wPRS was similar to that of the categorized model after stratification into low-risk, medium-risk, and high-risk tertiles (AUROC = 0.739, 95% CI: 0.668–0.810) (Figure 3A). Patients with high-risk scores had a higher chance of cognitive impairment than those with low-risk scores (HR = 18.22, 95% CI: 9.12–36.41; *p* = 2.1 × 10^−16^) (Figure 3B and Appendix A). Interestingly, the wPRS system was found to be the most effective predictor of developing incidental events of cognitive impairment in SAH patients with Hp2-1 (HR = 44.59, 95%f CI, 8.61–231.08; AUROC = 0.774, sensitivity = 0.333, specificity = 0.949) when compared to Hp2-2 (HR = 13.13, 95%f CI, 5.99–28.78; AUROC = 0.737, sensitivity = 0.345, specificity = 1.000). However, a subgroup analysis of Hp phenotypes was likely to be skewed due to small sample sizes, showing a high variance.

## 4. Discussion

Several studies have been conducted, mainly by testing clinical variables and radiological data, to identify modifiers related to cognition impairment following SAH. Rowland et al. reported that changes in brain volume 72 h after ictus were highly correlated with cognitive impairment in patients with SAH [20]. Increased brain volume with restricted water diffusion in the posterior cerebellum was significantly observed in patients. Persistent abnormalities in the volume of the posterior cerebellum 3 months post-SAH were present in patients with worse cognitive outcomes. In the acute phase after SAH, cognitive impairment was closely associated with acute hydrocephalus, cerebral infarction, and cerebrospinal fluid drainage over 2000 mL [6]. Ravnik et al. reported that frontal lobe damage was associated with a clear decline in visual memory, followed by a decline in verbal memory and executive functions, despite long-term favorable outcomes after SAH [21]. Compared to clinical and imaging studies, genetic studies on cognitive impairment in patients with SAH are relatively scarce. Moreover, the genetic studies conducted were mostly limited to simple candidate gene association analyses, such as *APOE*, insulin-like growth factor 1 (IGF-1), tumor necrosis factor-A (TNF-A), and interleukin-1A (IL-1A) and IL-1B [1]. Gallek et al. reported that the *APOE* ε4 allele, but not the *APOE* ε2 allele, was closely linked to poor functional outcomes within 6 months after SAH after adjusting for cerebral vasospasm [20]. Instead, the wild-type IGF-1 allele and non-wild type TNF-alpha allele were closely associated with cognitive impairment outcomes. A previous GWAS involving 245 Caucasian patients with SAHs reported that rs999662, located in chromosome 16q.13, was associated with a significantly increased risk of cerebral vasospasm diagnosed by transcranial Doppler [22]. In particular, carrying the TT variant of rs999662 was significantly associated with vasospasm. However, the study did not evaluate causal variants associated with cognitive impairment.

In this longitudinal GWAS, we identified five novel susceptible loci whose role in SAH or cognitive impairment was not well known in previous studies. *IRS2* tagging identified multiple variations within the 13q34 region encoding for the 180 kDa adapter protein, which is equivalent to the 4PS protein, and a substrate associated with the interleukin-4 receptor in myeloid cells [23]. The lack of *IRS2* modifies neurotransmitter release. The complete disruption of *IRS2* impaired the long-term potentiation of synaptic transmission in mice hippocampi [24]. Ochiai et al. also reported that the genetic deletion of *IRS2* suppressed amyloid-β accumulation in transgenic with AD mutant β-amyloid precursor protein overexpression [25]. However, the mechanism of cognitive impairment after SAH is thought to be vascular dementia, which has different clinical–pathological characteristics from those of AD. However, vascular dementia can also occur in diabetes mellitus, without accelerated β-amyloid deposition [26]. The expression of *IRS2* was also decreased in vascular dementia, as well as human metabolic traits such as diabetes mellitus and obesity. In addition, insulin resistance can lead to cognitive impairment and dementia states [27]. In particular, inflammation of the hypothalamus, with high numbers of insulin receptors, may impair regulation of the insulin signaling system. Lee et al. reported that SAH may cause hypothalamic injury [28]. In their study using diffusion tensor imaging, SAH patients exhibited significantly lower fractional anisotropy and higher apparent diffusion coefficient levels compared to those of the controls. Accordingly, the impairment of white matter integrity and increased water diffusion was more evident in the hypothalamus of patients with SAH. Our findings support the evidence indicating that neuroinflammation in the hypothalamus and impairment in insulin resistance regulation may pose a risk for cognitive impairment after SAH. Follow-up studies are needed to investigate the mechanism by which the sequence differences in *IRS2* cause cognitive impairment and whether the differences can be used as a therapeutic target in patients with SAH.

Population-based studies have developed genetic risk scores for specific populations using GWAS-driven loci to identify individuals at high risk for complex diseases such as coronary heart disease and ischemic stroke [29,30]. A PRS approach improves disease-risk predictability by computing the cumulative effect of multiple susceptibility loci [31]. Moreover, the models combining multiple genetic variants with conventional risk factors such as age, hypertension, obesity, and smoking can potentially enhance the predictive power for cerebrovascular diseases. In addition, cardiovascular risk factors, particularly hypertension, are associated with age-related cognitive impairment [32]. Therefore, follow-up studies are needed to construct a risk scoring system that can effectively predict an individual’s risk of cognitive impairment after SAH.

In this study, we identified cognition impairment-associated novel GWA signals and proposed a wPRS based on the five SNPs predicting cognitive impairment after SAH in a prospective cohort. Polygenic risk-informed cognitive impairment screening after SAH was likely to be more cost-effective than other alternatives, if targeted loci can well predict this disease in other independent groups replicatively, reducing the cost of additional screening tests for cognitive impairment. Furthermore, well-targeted (or mapped) markers (or loci) can contribute to achieving a time–cost-effective strategy by reducing unnecessary genotyping. Our developed model still requires a validation step and exhibits limitations that are yet to be resolved; thus, we are making efforts to increase the reliability for clinical utility, while replicating our model under other populations, as well as to discover disease coding modifiers correlating to current findings in exome sequencing data to effectively diagnose cognitive impairment following SAH [33].

While their application in clinical relevance may still be contentious, our PRS will clearly provide better information and potential benefit in the clinical research arena. A strong role of genetics or genomics in the disease etiology may even imply that patients or subjects with a high PRS benefit less from lifestyle changes, such as nutritional modification or smoking cessation, than those with a low PRS. Moreover, the advantages of earlier or more frequent screening using a reliable PRS-based panel will be able to reduce potential side effects such as life-threatening injuries occurring from radiation exposure, unnecessary screening, overdiagnosis, and neuro-degenerative distress after rupture symptoms [34].

Our study includes several limitations. First, the sample size was potentially underpowered to detect susceptible loci and polygenic variations regarding cognitive impairment after SAH because of rare frequent events of this impairment in patients with SAH. Thus, we plan to increase the number of patients with SAH in the follow-up study. In addition, risk estimation should be approached in various aspects at the individual- and population-level. Although we first reported longitudinal GWAS analyses for cognitive impairment following SAH, the inherent study concern of a relatively small sample size is a limitation. Second, an independent study should be requested to validate the variations and the wPRS models in regards to cognitive impairment after SAH over a longer time period. Third, the subgroup analysis of the Hp subtypes using wPRS did not approach sufficient statistical power due to a small sample size under reclassification. The wPRS system was especially effective in predicting cognitive impairment in patients with Hp2-1 whose risk of cognitive impairment was lower than those with Hp2-2. Although our findings were a novel approach to evaluate risk assessment of cognitive impairment after SAH in Hp subgroups, these results could be underpowered or could overestimate risks due to the relatively small sample size compared to those of other wPRS models for coronary artery disease or stroke [30,35]. SAH exhibits a lower incidence than coronary artery disease or stroke. Moreover, it is possible that cognitive impairment after SAH appears to be a very rare event during the follow-up period because most affected patients could be elderly and thus have died due to SAH. Accordingly, GWAS results based on a population-based cohort study should be performed in the future. 

Despite these limitations, our future study will include more patients by considering the collaboration between multi-centered hospitals via The First KSGAR Consortium (https://www.1ksgh.org/) to evaluate more reliable polygenic risks in traits such as the Hp subtypes. Fourth, our Cox proportional hazard failure time model is adapted to the retrospective epidemiological study. Some enrolled participants had their event time censored due to the end of our study, death, or some other competing event. Our characteristics of our study depend on the assumption that any censoring is independent of the cognitive impairment event after SAH during a 60-month (5-year) period. Nevertheless, our findings may provide useful information to be validated in additional independent or large-scale studies of patients with SAH. Above all, we aim to conduct a functional validation study to discover coding modifications or mechanisms related to the current findings by applying whole-exome sequencing, expression quantitative trait locus (eQTL) analyses, as well as a transcriptome-wide association study. Fifth, we defined cognitive impairment as an MMSE score lower than 27. This can be controversial, as MMSE scores lower than 24 have been suggested to indicate cognitive impairment [14]. Compared to AD, SAH-related cognitive impairment is more likely due to vascular dementia, and patients often deteriorate more rapidly than those with AD. Even when SAH patients seen in clinical practice complain that their cognitive impairment is getting worse, the MMSE result is often 24 or higher. MMSE scores less than 27 have been used to diagnose cognitive impairment following SAH [36]. These findings and measurements suggest that the current genetic risk score can be limited when considering clinical heterogeneity identifying patients with a high risk of SAH [37]. Nevertheless, efforts regarding risk assessment will be approached in various aspects in individual- and population-level studies to understand the etiology of human complex diseases or complications such as cognitive impairment and SAH [38].

In conclusion, this study implies a first investigation to identify modifiers of cognitive dysfunction after SAH, detecting five novel susceptible loci based on a prospective longitudinal GWAS. Although this work could not be replicated in other studies because, to the best our knowledge, there are no other studies of this kind available in previous research, it might provide examples of potential susceptibilities to cognitive impairment after brain vessel rupture. Polygenic risk assessment based on SNPs can be feasible in predicting cognitive impairment following SAH. Prospective studies including a larger number and/or independent studies including SAH patients will be required to identify the clinical relevance to our findings.

## Figures and Tables

**Figure 1 biomedicines-12-01387-f001:**
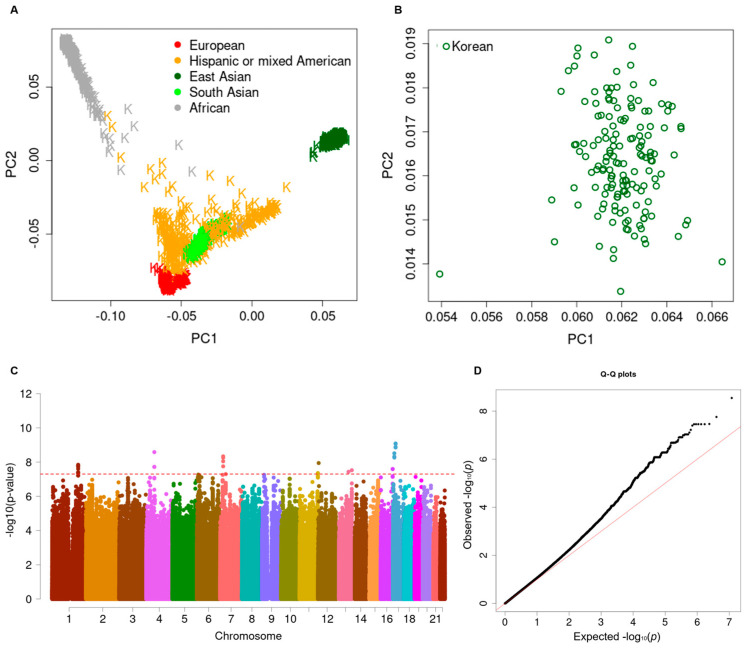
(**A**) A plot of the multidimensional scaled principal component (MDS PC) values in 153 Korean adults with subarachnoid hemorrhages (SAHs), plus 2504 subjects in the 1000 Genome (1KG) reference panel (Phase 3, version 5) and (**B**) a magnified distribution of MDS PC values in 153 patients with SAH. Genetic ancestry clusters between PC1 and PC2 show good dispersion of the patients in our cohort compared to 504 East-Asian subjects in the 1KG reference panel. K indicates 2504 subjects in the 1KG reference panel (the number of Europeans = 503, Hispanic or mixed Americans = 347, Africans = 661, East Asians = 504, and South Asians = 489). The dark green color indicates Korean adults, marked with an “H” in panel A. The green circle indicates the current study subjects in panel B. (**C**) Manhattan plot of the longitudinal genome-wide association study regarding the development incidental events of cognitive impairment following SAH. X- and Y-axes indicate the chromosome number (marked different colors) and -log10-transformed *p*-value, respectively. The red sub-line indicates the genome-wide significance threshold (*p* = 5 × 10^−8^). (**D**) The quantile–quantile plot of longitudinal genome-wide association study of the development of incidental events of cognitive impairment following SAH. The X- and Y-axes indicate the expected and observed scales of −log10-transformed *p*-value, respectively. The red solid line indicates a linear regression between observed and expected values.

**Figure 2 biomedicines-12-01387-f002:**
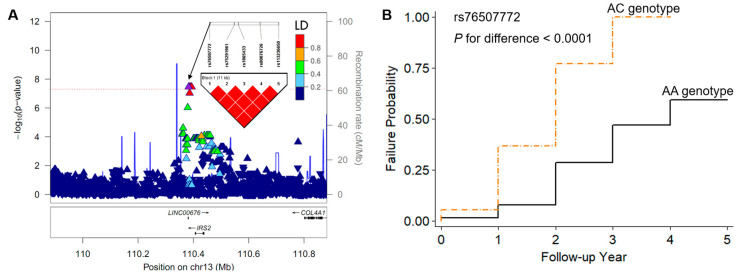
(**A**) A regional association plot of the *LINC00676-IRS2* region (13q34.3 chr13:109882882-110882882; rs76507772 SNP ± 500 kb) associated with cognitive impairment in patients with subarachnoid hemorrhages (SAHs). Triangles or reverse triangles indicate positive and negative effect sizes, respectively, and each color shows pairwise linkage disequilibrium. A purple up-pointing triangle indicates the top SNP of rs76507772. Other colors indicate pairwise linkage disequilibrium (LD, *r*^2^): navy, 0–0.2; green, 0.2–0.4; sky-blue, 0.4–0.6; orange, 0.6–0.8; red, 0.8–1. (**B**) The Kaplan–Meier survival curve, with failure probability, shows a genotype difference in the rs76507772 SNP for developing incidental events of cognitive impairment after SAH. Black solid line, non-risk homozygous genotype; orange dashed line, heterozygous genotype. The *p*-value indicates a difference between genotypes.

**Figure 3 biomedicines-12-01387-f003:**
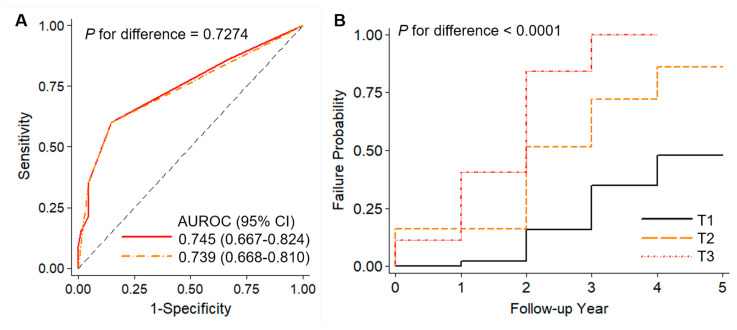
(**A**) Area under the receiver-operating curves (AUROCs) for weighted polygenic risk score (wPRS) models for developing cognitive impairment (CI) following subarachnoid hemorrhage (SAH). Red solid and orange dash-dot lines indicate the wPRS model accuracy and stratification of the model into low-risk (T1), medium-risk (T2), and high-risk (T3) tertiles, respectively. (**B**) The Kaplan–Meier survival curve, with failure probability, shows the development of incidental events of cognitive impairment after SAH during follow-up, based on three polygenic risk groups. Lines indicate the low-risk (black solid line), medium-risk (orange dashed line), and high-risk (red dotted line) groups. The *p*-value for differences between the risk groups is provided.

**Table 1 biomedicines-12-01387-t001:** Baseline characteristics of the study participants.

Variables ^a^	Cognitive Impairment (N = 65)	No Cognitive Impairment (N = 88)	*p*
Clinical factors			
Female	44 (67.7%)	53 (60.2%)	0.698
Age, years	64.2 ± 12.3	57.5 ± 11.0	<0.001
Hypertension	38 (58.5%)	34 (38.6%)	0.002
Diabetes mellitus	8 (12.3%)	7 (8.0%)	0.085
Hyperlipidemia	5 (7.7%)	11 (12.5%)	0.481
Smoking	8 (12.3%)	12 (13.6%)	0.873
SAH-related variables			
Aneurysm size (mm)	4.8 ± 1.3	4.6 ± 1.2	0.589
Anterior circulation aneurysm	55 (84.6%)	78 (88.6%)	0.640
Hunt and Hess grade IV and V	25 (38.4%)	26 (29.5%)	0.247
Fisher grade III and IV	32 (49.2%)	35 (39.7%)	0.244
Delayed cerebral ischemia	17 (26.2%)	21 (23.9%)	0.746
Hydrocephalus	7 (10.8%)	5 (5.7%)	0.247
Treatment methods			
Coil embolization	61 (93.8%)	78 (88.6%)	0.269

^a^ Data are described with the numbers of subjects (percentages), for discrete variables, and mean ± standard deviation, for continuous variables.

**Table 2 biomedicines-12-01387-t002:** Longitudinal genome-wide association study of cognitive impairment following subarachnoid hemorrhage.

Gene	Chr.	Region	SNP	BP	M/m ^a^	MAF	HWE *p*	HR ^b^	95% CI ^b^	*p* ^b^
*PDCD6IP-LOC101928135*	3p22.3	intergenic	rs138753053	33961435	G/A	0.013	1	28.33	8.64–92.92	3.4 × 10^−8^
*LINC00499*	4q28.3	ncRNA	rs56823384	139255670	T/C	0.026	1	12.47	5.42–28.67	2.8 × 10^−9^
*CASC15*	6p22.3	ncRNA	rs145397166	22067476	C/G	0.026	1	11.16	4.82–25.83	1.7 × 10^−8^
*LPL-SLC18A1*	8p21.3	intergenic	rs10503670	19985382	G/A	0.490	0.178	2.88	1.97–4.20	4.0 × 10^−8^
*IRS2*	13q34	near UTR-3	rs76507772	110382882	A/C	0.062	1	5.99	3.17–11.32	3.5 × 10^−8^

BP, base-pair position; Chr, chromosome; CI, confidence interval; HR, hazard ratio; HWE *p*, Hardy–Weinberg equilibrium *p*-value; MAF, minor allele frequency; SNP, single nucleotide polymorphism. ^a^ M/m indicates a major and minor allele, respectively. ^b^ HR, 95% CI, and *p*-value were estimated using a multivariate Cox proportional hazard regression test in a longitudinal genome-wide association study. All SNPs exhibit above 0.7 of *R*^2^ (imputation score), determined by the Minimac4 program.

**Table 3 biomedicines-12-01387-t003:** Multiple Cox proportional hazard regression models of five single nucleotide polymorphisms (SNPs) with cognitive impairment, adjusting for potential bias.

rs138753053		rs56823384		rs145397166		rs10503670		rs76507772	
HR (95% CI) ^a^	*p* ^a^	HR (95% CI) ^a^	*p* ^a^	HR (95% CI) ^a^	*p* ^a^	HR (95% CI) ^a^	*p* ^a^	HR (95% CI) ^a^	*p* ^a^
Adjusted model 1: sex + age + targeted SNP									
28.33 (8.64–92.92)	3.4 × 10^−8^	12.47 (5.42–28.67)	2.8 × 10^−9^	11.16 (4.82–25.83)	1.7 × 10^−8^	2.88 (1.97–4.2)	4.0 × 10^−8^	5.99 (3.17–11.32)	3.5 × 10^−8^
Adjusted model 2: sex + age + hypertension + diabetes + targeted SNP									
23.48 (6.92–79.67)	4.1×10^−7^	10.68 (4.5–25.38)	8.1 × 10^−8^	10.16 (4.24–24.34)	2.0 × 10^−7^	2.82 (1.93–4.13)	8.3 × 10^−8^	5.69 (2.99–10.84)	1.3 × 10^−7^
Adjusted model 3: sex + age + hypertension + diabetes + hyperlipidemia + smoking + targeted SNP									
19.10 (5.49–66.39)	3.5×10^−6^	11.17 (4.52–27.61)	1.7 × 10^−7^	9.60 (3.86–23.90)	2.1 × 10^−6^	2.78 (1.87–4.12)	3.8 × 10^−7^	5.35 (2.77–10.32)	5.8 × 10^−7^

^a^ Hazard ratio (HR), confidence interval (CI), and *p*-value were estimated from multivariate Cox proportional hazard regression models after additional adjustment for covariates of each model, plus four genetic ancestry (principal component analysis, PCA) values. The result of the PCA values were hidden in each model because they showed extremely high or low statistics (e.g., HR), such as 5.53 × 10^47^ or 8.63 × 10^−9^.

## Data Availability

All relevant data, including summary statistics, are provided within the paper and its Appendix A. The original raw genotype data contains potentially sensitive data, including patient’s genomic profile, and access to this data is restricted by the Ethics Committee and Institutional Review Board (IRB) of Hallym University Chuncheon Scared Hospital (https://chuncheon.hallym.or.kr/irb/index.asp). Interested researchers can submit “The Agreement Form of Controlled Data Usage” to the Ethics Committee and IRB to request access to the data.

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
