# Peer review of "Longitudinal Genome-Wide Association Study of Cognitive Impairment after Subarachnoid Hemorrhage"

_biomedicines, 2024, doi:10.3390/biomedicines12071387_

Round 1
Reviewer 1 Report
Comments and Suggestions for Authors
The authors submitted a research paper in which they reported the results of the longitudinal genome-wide association study for the first time to identify genetic modifications associated with cognitive defection after subarachnoid hemorrhage. They established genotype difference in the rs76507772 SNP for developing incidental events of the cognitive impairment after SAH. Yet, they identified 5 SNPs including rs138753053, rs56823384, rs145397166, rs10503670, and rs76507772, which were significantly associated with cognitive impairment in patients with APOE ɛ3/ɛ3. The findings seem to be impressive and having serious clinical significance. However, I would like to make some comments to discuss them.
1. Although GWAS technology allows to identify loci in individuals at high risk, there is a problematic issue whether genetic predictive score is sufficient tool for reclassification of the CVD and cerebrovascular disease individuals. Please, discuss the issue taking into consideration current clinical guideline for CVD prevention of ESC
2. Please, link polygenic risk assessment with additional economic burden and rpovid your opinion on it.
Author Response
Manuscript ID: biomedicines-3043896
“Longitudinal Genome-wide Association Study of Cognitive Impairment After Subarachnoid Hemorrhage” Thank you for your kind advice on this paper. We have submitted a revised manuscript informed by the Reviewers’ comments, highlighting a revised sentences or tables with a gray color, describing strikethroughs on the deleted sentences or words, and replacing a reference marking of (number) into [number] in the text with reformatting and renumbering several references.
Please read a attached letter

Reviewer 2 Report
Comments and Suggestions for Authors
In this study, the researchers investigated genetic factors associated with cognitive impairment following subarachnoid hemorrhage (SAH). Conducted on 153 SAH patients, the study identified five significant genetic loci linked to cognitive decline. A weighted polygenic risk score based on these loci predicted cognitive impairment with an AUROC of 0.745. This study may suggest new genetic markers that could help predict and manage cognitive outcomes in SAH patients.
1. The tables are very difficult to read. They should be reformatted.
2. The results of APOE ɛ alleles are contradictory throughout the manuscript and should be revised.
3. The discussion is confusing and loose. Most of it seems to me irrelevant.
Comments on the Quality of English Language
The language of the manuscript should be revised.
Author Response
Manuscript ID: biomedicines-3043896
“Longitudinal Genome-wide Association Study of Cognitive Impairment After Subarachnoid Hemorrhage”
Thank you for your kind advice on this paper. We have submitted a revised manuscript informed by the Reviewers’ comments, highlighting a revised sentences or tables with a gray color, describing strikethroughs on the deleted sentences or words, and replacing a reference marking of (number) into [number] in the text with reformatting and renumbering several references.
Please find, confirm, and read a attached letter.

Round 2
Reviewer 2 Report
Comments and Suggestions for Authors
The manuscript has been significantly improved by the authors.